# Protein Arginine *N*-methyltransferases 5 and 7 Promote HIV-1 Production

**DOI:** 10.3390/v12030355

**Published:** 2020-03-23

**Authors:** Hironobu Murakami, Takehiro Suzuki, Kiyoto Tsuchiya, Hiroyuki Gatanaga, Manabu Taura, Eriko Kudo, Seiji Okada, Masami Takei, Kazumichi Kuroda, Tatsuo Yamamoto, Kyoji Hagiwara, Naoshi Dohmae, Yoko Aida

**Affiliations:** 1Viral Infectious Disease Unit, RIKEN, 2-1 Hirosawa, Wako, Saitama 351-0198, Japan; h-murakami@azabu-u.ac.jp (H.M.); takei.masami@nihon-u.ac.jp (M.T.); kyoji@riken.jp (K.H.); 2Laboratory of Animal Health II, School of Veterinary Medicine, Azabu University, 1-17-71 Fuchinobe, Chuo-ku, Sagamihara, Kanagawa 252-5201, Japan; 3Biomolecular Characterization Unit, RIKEN CSRS, 2-1 Hirosawa, Wako, Saitama 351-0198, Japan; takehirosuzuki@riken.jp; 4AIDS Clinical Center, National Center for Global Health and Medicine, 1-21-1 Toyama, Shinjuku-ku, Tokyo 162-8655, Japan; ktsuchiy@acc.ncgm.go.jp (K.T.); higatana@acc.ncgm.go.jp (H.G.); dohmae@riken.jp (N.D.); 5Division of Hematopoiesis, Joint Research Center for Human Retrovirus Infection, Kumamoto University, 2-2-1 Honjo, Chuo-ku, Kumamoto 860-0811, Japan; manabu.taura@yale.edu (M.T.); erikokudo0930@gmail.com (E.K.); okadas@kumamoto-u.ac.jp (S.O.); 6Nakamura Laboratory, Baton Zone Program, RIKEN Cluster for Science, Technology and Innovation Hub, 2-1 Hirosawa, Wako, Saitama 351-0198, Japan; 7Nihon University School of Medicine, 30-1 Oyaguchi, Itabashi, Tokyo 173-8610, Japan; kuroda.kazumichi@nihon-u.ac.jp (K.K.); yamamoto.tatsuo@nihon-u.ac.jp (T.Y.)

**Keywords:** HIV-1, Vpr, PRMT5, PRMT7, Vpr-binding protein, virus production, stability, macrophage, pathogenesis

## Abstract

Current therapies for human immunodeficiency virus type 1 (HIV-1) do not completely eliminate viral reservoirs in cells, such as macrophages. The HIV-1 accessory protein viral protein R (Vpr) promotes virus production in macrophages, and the maintenance of Vpr is essential for HIV-1 replication in these reservoir cells. We identified two novel Vpr-binding proteins, i.e., protein arginine *N*-methyltransferases (PRMTs) 5 and 7, using human monocyte-derived macrophages (MDMs). Both proteins found to be important for prevention of Vpr degradation by the proteasome; in the context of PRMT5 and PRMT7 knockdowns, degradation of Vpr could be prevented using a proteasome inhibitor. In MDMs infected with a wild-type strain, knockdown of PRMT5/PRMT7 and low expression of PRMT5 resulted in inefficient virus production like Vpr-deficient strain infections. Thus, our findings suggest that PRMT5 and PRMT7 support HIV-1 replication via maintenance of Vpr protein stability.

## 1. Introduction

The viral load (i.e., the number of copies) of human immunodeficiency virus type 1 (HIV-1) in the bloodstream is closely related to the development of acquired immunodeficiency syndrome (AIDS) [1,2]. Rapid progressors, who typically progress to AIDS a few years after infection in the absence of highly active antiretroviral therapy (HAART), maintain a high viral load [3], whereas both long-term non-progressors and controllers maintain a low viral load without HAART [4]. Thus, HAART, which can control the viral load, is an effective therapy [5]. However, HAART does not completely eliminate viral reservoirs, such as myeloid-lineage cells, monocytes, and macrophages [6]. Macrophages are an early cellular target for HIV-1 infection; indeed, transmission between humans is caused by macrophage-tropic virus rather than CD4^+^ T cell-tropic virus [7]. Although, CD4^+^ T cells infected with T cell-tropic HIV-1 are the main source of virus production [7,8,9], macrophage-tropic HIV-1 is detected during rebound viremia caused by discontinuation of HAART [10,11,12]. In addition, HIV-1-infected macrophages contribute to viremia [13]. Therefore, an identification of the mechanisms underlying virus production in macrophages is essential for understanding HIV-1 pathogenesis.

The HIV-1 accessory protein viral protein R (Vpr) is related to HIV-1 pathogenesis and has multiple functions. For example, Vpr induces cell cycle arrest at G_2_ phase [14,15], promotes nuclear import of the viral pre-integration complex in non-dividing cells [16,17,18,19,20,21], regulates apoptosis [22,23,24,25], activates the transcription of the HIV-1 long terminal repeat [26,27], selectively inhibits cellular pre-mRNA splicing both, in vivo and in vitro [28,29], and antagonizes DNA repair [30]. Moreover, Vpr has the potential to increase virus production in cells of the monocyte-macrophage lineage and in a subtype of CD4^+^ T cells [31,32,33,34,35,36,37]. Vpr carries out its functions through interactions with various host factors, such as DDB1- and CUL4-associated factor 1 (DCAF1), spliceosome-associated protein 145 (SAP145), p300, synthetic lethal of unknown (X) function 4 (SLX4), importin α, and mini-chromosome maintenance protein10 (MCM10) [18,19,27,29,38,39,40,41]. Although, Vpr function is important for HIV-1 pathogenesis in macrophages, the specific mechanisms of Vpr in this context are still unclear. 

In this study, we examined novel Vpr-binding proteins in macrophages and from that examination the protein arginine *N*-methyltransferases 5 and 7 (PRMT5 and PRMT7) were identified. The PRMT family consists of ten PRMT proteins (PRMT 1–10) that have been identified to date, and all PRMT proteins, except PRMT8, are expressed in various tissues. However, PRMT expression in macrophages has not been investigated [42]. In the current study, we investigated the role of PRMT5 and PRMT7 in Vpr-mediated virus production in macrophages.

## 2. Materials and Methods

### 2.1. Cells and Culture Conditions

Monocyte-derived macrophages (MDMs) were differentiated from monocytes isolated from human peripheral blood mononuclear cells (PBMCs) using CD14 MicroBeads (Miltenyi Biotec, Bergisch Gladbach, Germany). The human PBMCs were collected from eight different healthy donors (Donors 1–8). The monocytes were differentiated by treatment with human macrophage colony-stimulating factor (M-CSF; PeproTech, London, UK) for 1 week. The MDMs were maintained in RPMI 1640 (Thermo Fisher Scientific, Waltham, MA, USA) supplemented with 10% fetal bovine serum (FBS) (Promega, Madison, WI, USA), 5% human serum from human male AB plasma (Sigma Aldrich, St. Louis, MO, USA), 0.2 % GlutaMax (Thermo Fisher Scientific), and 10 ng/mL M-CSF. HeLa and 293T cells were maintained in Dulbecco’s modified Eagle’s medium (Thermo Fisher Scientific) supplemented with 10% FBS (Promega). All cells used in this study were cultured at 37 °C in a humidified atmosphere of 5% CO_2_.

### 2.2. Expression Vectors

The expression vector pCAGGS [43], encoding FLAG-mRFP-FLAG-Vpr or FLAG-mRFP, has been described previously [44]. PRMT genes, except PRMT8, were amplified by polymerase chain reaction (PCR) from cDNA synthesized from HeLa cells. The PRMT8 gene was amplified by PCR from cDNA synthesized from human brain tissue (Clontech, Mountain View, CA, USA). The resulting PCR products were cloned into pGEX-6p-3 (GE Healthcare, Piscataway, NJ, USA) and pCAGGS vectors. The bicistronic vector HA-Vpr-IRES-ZsGreen1-pCAGGS, which contained HA-tagged Vpr plus IRES and ZsGreen1, was then constructed. 

### 2.3. Identification of Vpr-Binding Proteins

HeLa cells were transfected with FLAG-mRFP (mRFP)- or FLAG-mRFP-FLAG-Vpr (mRFP-Vpr)-pCAGGS expression vectors using FuGENE HD (Promega, Madison, WI, USA), as described previously [44]. Cells were harvested 2 days post-transfection and lysed in CelLytic M buffer (Sigma Aldrich, St. Louis, MO, USA). The mRFP and mRFP-Vpr were immunoprecipitated using anti-FLAG M2-conjugated agarose beads (Sigma Aldrich) and washed with wash buffer (10 mM Tris-HCl (pH 7.8), 150 mM NaCl, 1% Triton X-100). 

Monocytes from healthy donors were differentiated into MDMs, as described previously [18,44]. MDMs were suspended in binding buffer (10 mM Tris-HCl (pH 7.8), 1 mM ethylenediaminetetraacetic acid) and lysed with four freeze-thaw cycles. The MDM lysates were incubated with purified mRFP-Vpr conjugated to agarose beads at 4 °C for 4 h and then washed with binding buffer. Protein complexes were eluted by heat denaturing and then solubilized in Laemmli sample buffer. The proteins were analyzed by electrophoresis on 10% sodium dodecyl sulfate (SDS) polyacrylamide gels and visualized by Coomassie brilliant blue (CBB) staining. The CBB-stained bands were excised, digested with trypsin, and analyzed using an Ultraflex matrix-assisted laser desorption ionization time-of-flight (MALDI-TOF) mass spectrometer (Bruker Daltonics, Bremen, Germany). Peptide mass fingerprinting was performed by screening each peak list against the NCBI nr database using the MASCOT program (Matrix Science, London, UK). 

### 2.4. Glutathione-S-Transferase (GST) Pull-Down Assay

*Escherichia coli* strain BL21 cells were transformed with GST, GST-tagged Vpr (GST-Vpr), or GST-HA-tagged PRMT1–10 (GST-PRMT) expression vectors. The expressed GST, GST-Vpr, and GST-PRMT proteins were purified using glutathione Sepharose beads (GE Healthcare, Piscataway, NJ, USA). GST was removed from GST-PRMTs using PreScission protease (GE Healthcare), according to the manufacturer’s instructions. Each GST-PRMT was then incubated with GST or GST-Vpr for 2 h at 4 °C. PRMT proteins were detected by immunoblotting. 

### 2.5. Immunoblotting

Complexes obtained from GST-pull down/co-immunoprecipitation assay and plasmid-transfected cells washed with phosphate-buffered saline (PBS) were resuspended in Laemmli sample buffer and heat denatured. Proteins were separated on 20% SDS polyacrylamide gels and transferred to polyvinylidene difluoride membranes (Immobilon; Millipore, Burlington, MA, USA). The blotted membranes were blocked with PBS containing 0.05% Tween-20 (PBST) and 5% skim milk at room temperature for 1 h. After washing with PBST three times, the membranes were incubated with anti-HA monoclonal antibodies (MAbs; Sigma Aldrich), anti-PRMT2 (Santa Cruz Biotechnology, Dallas, TX, USA), anti-PRMT5 (Santa Cruz Biotechnology), anti-PRMT7 (Abcam, Cambridge, UK), anti-PRMT9 (Abcam), or anti-β-actin polyclonal antibodies (MBL, Nagoya, Japan) diluted with PBST containing 3% skim milk at 4 °C for 16–18 h. After washing three times with PBST, the membranes were incubated with horseradish peroxidase (HRP)-conjugated anti-mouse IgG (Jackson immunoresearch, West Grove, PA, USA) or HRP-conjugated anti-rabbit IgG (Jackson immunoresearch) at room temperature for 1 h. Each antibody was diluted with PBST containing 3% skim milk. After washing the membrane three times with PBST, proteins were visualized using Pierce ECL Western Blotting Substrate (Thermo Fisher Scientific, Waltham, MA, USA), according to the manufacturer’s instructions. 

### 2.6. Co-Immunoprecipitation

HeLa cells were transfected with HA- or HA-Vpr-expressing vectors using FuGENE HD (Promega, Madison, WI, USA). After 2 days incubation, the cells were lysed in CelLytic M (Sigma Aldrich), and the lysates were incubated with anti-HA-conjugated agarose beads (Sigma Aldrich) for 3 h at 4 °C. The complexes were then washed three times with Triton X-100-free wash buffer (10 mM Tris-HCl (pH 7.8), 150 mM NaCl) and analyzed by immunoblotting, as previously described [18]. 

### 2.7. siRNA and Transfection

Stealth RNAi siRNA Duplex Oligonucleotides and control siRNA (siRandom) were purchased from Invitrogen (Carlsbad, CA, USA). The siRNA target sequences were as follows: siPRMT2, 5′-CAGAACGGCUUUGCUGACAUCAUCA-3′; siPRMT5, 5′-CACUGAUGGACAAUCUGGAAUCUCA-3′; siPRMT7, 5′-GAGCAGGUGUUUACAGUCGAGAGUU-3′; and siPRMT9, 5′-GGAAAGAGAGUUUCCAGCAGUUGUA -3′. MDMs from a healthy donor were transfected with siRNA using Lipofectamine 2000 at 24 or 48 h after differentiation from monocytes. HeLa cells were cotransfected with siRNAs and plasmids using Lipofectamine 2000 (Invitrogen, Carlsbad, CA, USA), according to the manufacturer’s instructions.

### 2.8. Quantitative PCR (qPCR)

RNA was isolated from the MDMs using a RNeasy kit (Qiagen, Valencia, CA, USA). The isolated RNA was used as a template to synthesize cDNA with a High-Capacity cDNA Reverse Transcription Kit (Thermo Fisher Scientific, Waltham, MA, USA). PRMT5, PRMT7, and β-actin expression were analyzed by qPCR using a 7500 Real-Time PCR system (Applied Biosystems, Foster City, CA, USA). The expression level of PRMT5 and PRMT7 were normalized to that of β-actin.

### 2.9. Viruses and Infection

We used infectious molecular clones (HIV-1 pNF462 WT and pNF462 ΔR), encoding wild-type Vpr (NF462 WT) [45] and mutated Vpr (Vpr-negative mutant, NF462 ΔR), respectively [18]. The virus strains NF462 WT and NF462 ΔR were produced from pNF462 WT- and ΔR-transfected 293T cells, as previously described [44,46]. HIV-1 titers were measured using an anti-p24 enzyme-linked immunosorbent assay (ELISA) kit (Ryukyu Immunology, Okinawa, Japan). MDMs were incubated with a specific titer (p24; 1 ng/mL) for 2 h. After incubation, MDMs were washed with RPMI 1640 three times, and fresh growth medium was added. MDM media were collected at 0, 4, 8, and 12 days post-infection. After each collection, the MDMs were three times washed with RPMI 1640, and fresh growth medium was added. The collected media were used for measuring virus production using p24 ELISA. 

### 2.10. Statistical Analysis

Paired and unpaired Student’s t tests and one-way analysis of variance were performed using the statistical software R version 3.3.3 [47]. The results with *p* values of less than 0.05 were considered significant.

## 3. Results

### 3.1. Identification of Vpr-Binding Protein Derived from Human MDMs

To identify novel Vpr-interacting host factors in primary macrophages, we performed binding assays using Vpr protein purified from HeLa cells and lysates of human MDMs derived from Donor 1. Briefly, FLAG-tagged Vpr protein was purified from HeLa cells and incubated with MDM lysate. The resulting Vpr protein complexes were immunoprecipitated and resolved by SDS-polyacrylamide gel electrophoresis (PAGE; Figure 1a). The immunoprecipitated bands were analyzed by MALDI-TOF mass spectrometry (MS), and the peaks from a single protein of approximately 70 kDa were measured (Figure 1b). Although, several proteins were found based on the peaks, PRMT5 protein was identified as the most likely, with 19 peptide matches leading to 21% sequence coverage (Figure 1c). Thus, PRMT5 was identified as a Vpr-binding protein derived from human MDMs.

### 3.2. Four PRMT Family Proteins Bind to Vpr

To directly assess binding between PRMT5 and Vpr, we performed a GST pull-down assay using purified recombinant GST-Vpr, PRMT5, and a GST-only negative control. We found that PRMT5 was able to bind directly to GST-Vpr (Figure 2a).

Next, we investigated Vpr binding by other PRMT family members, which share common PRMT motifs [48]. We performed GST pull-down assays using purified recombinant HA-tagged PRMTs and found that PRMT2, PRMT7, and PRMT9 could bind directly to GST-Vpr (Figure 2b).

To address whether Vpr could bind to endogenous PRMT2, PRMT5, PRMT7, and PRMT9, HeLa cells were transfected with HA-tagged Vpr (HA-Vpr) and tested by co-immunoprecipitation. We found that HA-Vpr could specifically bind to endogenous PRMT2, PRMT5, PRMT7, and PRMT9 (Figure 2c).

### 3.3. Vpr Expression Affected by PRMT5 and PRMT7

A recent report hypothesized that Vpr degrades host factors [30]. Thus, we examined whether PRMT2, PRMT5, PRMT7, and PRMT9 were degraded by Vpr. Our results showed that these PRMTs, binding to Vpr, were not downregulated in the context of HA-Vpr overexpression (Appendix A). Thus, Vpr could bind to but not degrade these PRMTs.

PRMT family proteins recognize RGG/RG, PGM-rich, and RXR (R: arginine, G: glycine, P: proline, M: methionine, and X: any amino acid) motifs [49]. Vpr protein harbors an arginine-rich sequence at the C-terminal, and the amino acid sequence of Vpr from positions 85 to 90 (RQRRAR) encodes two RXR motifs. We hypothesized that these PRMTs affected Vpr function through methylation of the arginine-rich region of Vpr. Thus, we analyzed methylation sites using MS analysis. We found that Vpr was not methylated (Appendix A); therefore, PRMT2, PRMT5, PRMT7, and PRMT9 do not methylate Vpr, despite binding to it.

To further investigate the roles of PRMTs in Vpr function, we utilized siRNA-knockdown experiments. HeLa cells were transfected with siRNAs against PRMT2, PRMT5, PRMT7, or PRMT9 together with the bicistronic vector HA-Vpr-IRES-ZsGreen1-pCAGGS. PRMT knockdown was confirmed by immunoblotting at 48 h post-transfection. Interestingly, knockdown of PRMT5 or PRMT7 downregulated HA-Vpr in a concentration-dependent manner, but did not affect the expression of ZsGreen1. In contrast, knockdown of PRMT2 and PRMT9 did not affect the expression of HA-Vpr or ZsGreen1 (Figure 3a). As knockdown of PRMT5 or PRMT7 alone decreased HA-Vpr expression, we next analyzed Vpr expression in the context of double knockdown of PRMT5 and PRMT7. The double knockdown resulted in greater downregulation of HA-Vpr than knockdown of either PRMT5 or PRMT7 alone (Figure 3b). These results suggest that PRMT5 and PRMT7 affected Vpr expression both independently and synergistically.

### 3.4. Inhibition of Proteasome-Dependent Vpr Degradation by Two Vpr-Binding PRMTs

To investigate the stabilization of HA-Vpr by interaction with PRMT5 and PRMT7, we examined HA-Vpr expression by knockdown of PRMT5 and PRMT7 in the presence of the proteasome inhibitor lactacystin (LC) and cathepsin inhibitor E-64d. Briefly, HeLa cells were cotransfected with HA/HA-Vpr-IRES-ZsGreen1-pCAGGS and siRNAs against PRMT5 and/or PRMT7. After 48 h post-transfection, the cells were treated with LC or E-64d for 6 h. HA-Vpr expression was unchanged in the presence of LC in PRMT5- and PRMT7-knockdown cells (Figure 3c), indicating that downregulation of Vpr expression after PRMT5 and/or PRMT7 knockdown was rescued by LC. In contrast, E-64d treatment gave results like those of the negative control, which were HA-Vpr expression was reduced by knockdown of PRMT5 and PRMT7 (Figure 3d). These results showed that reduction of HA-Vpr protein expression by knockdown of PRMT5 and PRMT7 was caused by proteasomal degradation. 

### 3.5. Knockdown of Vpr-Binding PRMTs in MDMs Inhibited Efficient Virus Production

Vpr functions to promote virus production in cells of the monocyte-macrophage lineage [18,31,37]. We hypothesized that, because the expression level of PRMT5 and PRMT7 affected Vpr levels, HIV-1 production would also be affected by these proteins in macrophages. To test this hypothesis, we examined virus production following knockdown of PRMT5 and PRMT7 in human MDMs from Donor 2. Briefly, siPRMT5- and/or siPRMT7-transfected MDMs were infected with NF462 WT or NF462 ΔR strain and cultured for 12 days. Virus production was measured using p24 ELISA at 4, 8, and 12 days post-infection. To confirm the expression levels of PRMT5 and PRMT7, we performed immunoblotting. Unfortunately, these proteins could not reliably be detected because of the small sample volumes obtained from the HIV-1-infected MDMs. Therefore, we measured PRMT5 and PRMT7 expression levels by qPCR as described by Demetriadou et al [50]. The results showed that knockdown was still maintained at 12 days post-infection (Figure 4a). The NF462 WT strain showed significantly higher virus titer than the NF462 ΔR strain in cells transfected with siRandom at 12 days post-infection, when the expression levels of PRMT5 and PRMT7 were similar (Figure 4a, lanes 1 and 5). As hypothesized, the production of NF462 WT virus was inhibited by knockdown of PRMT5 and/or PRMT7. There were no significant differences in virus production between the NF462 WT and NF462 ΔR strains in the context of PRMT5 and/or PRMT7 knockdown at 12 days post-infection (Figure 4b). These results showed that low expression of PRMT5 and PRMT7 reduced the productivity of the NF462 WT strains to equal that of the NF462 ΔR strain, suggesting that PRMT5 and PRMT7 may promote efficient virus production in macrophages by stabilizing Vpr. 

### 3.6. Low Expression of PRMT5 Affected Wild-Type HIV-1 Replication in MDMs

Next, we examined whether low expression levels of PRMT5 and PRMT7 resulted in the same virus productivity in wild-type and Vpr-deficient strains in cells of the monocyte-macrophage lineage. As the expression of PRMT family proteins in macrophages had not been previously described, we first analyzed the expression of *PRMT5* and *PRMT7* in MDMs isolated from six individual donors (Donors 3–8) using qPCR. We discovered *PRMT5* expression differed among donors (Figure 5a); *PRMT5* mRNA expression was significantly lower in Donors 6, 7, and 8 than in Donors 3, 4, and 5 which had similar levels of expression. The mRNA expression level of *PRMT7* did not differ significantly among the six donors (Figure 5b).

To examine whether the expression level of PRMT5 influenced production of the NF462 WT strain, but not the NF462 ΔR strain, MDMs from the same six healthy donors (Donors 3–8) were infected with the NF462 WT or NF462 ΔR strain. Virus production was measured using p24 ELISA at 0, 4, 8, and 12 days post-infection. Virus production in MDMs from Donor 3 was dramatically higher than from other donors (Donors 4–8; Appendix A), suggesting that differences in infectious efficiency or viral productivity could be attributed to different donor-derived MDMs. Despite these variations in viral production levels, NF462 WT strain produced significantly higher titer than NF462 ΔR strain in MDMs carrying high PRMT5 expression (Donor 3, 4, and 5). In MDMs from Donors 6, 7, and 8, which exhibited low PRMT5 expression, there were no significant differences between production of the NF462 WT and the NF462 ΔR strains. This indicates that low PRMT5 expression reduced production of the wild-type strain to the level of the Vpr-deficient strain. This data suggests that the expression level of PRMT5 affected virus productivity through increased Vpr protein stability, although differences in the virus productivity between individuals were not affected by Vpr expression levels.

## 4. Discussion

In this study, we showed that PRMT5 and PRMT7 promoted HIV-1 production in primary macrophages by mediating the stability of Vpr by protecting it from proteasomal degradation. Although Vpr can degrade host factors, such as helicase-like transcription factor [30], the Vpr-binding proteins identified in this study, PRMT5 and PRMT7, were related to the stability of Vpr itself. In addition, our data showed that the virus production ability of the wild-type strain, with PRMT5/PRMT7 knockdowns, corresponded to that of the Vpr-deficient strain. We demonstrated that Vpr protein levels were not sufficiently maintained when PRMT5 and PRMT7 expression is low. MDMs with innately low PRMT5 expression showed similar levels of virus production when infected with either the wild-type or the Vpr-deficient strain. By contrast, the Vpr-deficient strain showed low virus production compared to the wild-type virus when infected into MDMs with high PRMT5 expression. The virus production levels in knockdown and low PRMT5-expression MDMs were similar, implying that the low PRMT5 expression lowered the virus production of the wild-type strain to that of Vpr-deficient strain because of degradation of Vpr. Vpr function is important in macrophages [36,51], and the improved Vpr stability by interaction with PRMT5 and PRMT7 supports higher virus production.

Previous studies have demonstrated the expression of Vpr-binding proteins, including DCAF1, SAP145, p300, SLX4, importin α, and MCM10 in cells other than MDMs [18,19,27,29,38,39,40,41]. In this study, we identified the novel Vpr-binding proteins PRMT5 and PRMT7. The results implied that the interactions between Vpr and host factors depended on the gene expression pattern in each cell. It was thought that the interaction pattern of Vpr differed among HIV-1-infected cells, and Vpr exerted distinct functions in each cell. The present study showed that the natural expression level of PRMT5 would show variations among MDMs from each individual although the virus production in MDMs with high PRMT5 expression was not same level. The observation showed that the variations in PRMT5 expression levels could not be attributed to individual differences in HIV-1 susceptibility, such as virus productivities in Donor 3 versus Donors 4–8. Although, differences in virus production between wild-type and Vpr-deficient infections could be clearly observed in MDMs from Donor 3, 4, and 5 compared to those from Donors 6, 7, and 8. The data suggest that the PRMT5 expression level does not correlate to high virus production but high PRMT5 expression stabilizes Vpr which in turn increases virus production.

In this study, we evaluated how PRMT5 and PRMT7 affected Vpr. Our results showed that, both proteins protected Vpr from proteasomal degradation, as demonstrated by recovery of Vpr expression following treatment with a proteasomal inhibitor in siRNA-transfected cells. A previous study demonstrated that PRMT5-mediated methylation inhibits proteasomal degradation [52], and other previous studies demonstrated that a methylation inhibitor decreased viral production [53,54]. However, in our study, Vpr was not methylated, as demonstrated by MS analysis. Therefore, further study is needed into mechanism of how PRMT5 and PRMT7 binding confers stability on Vpr. 

Previous studies demonstrated that degradation of host factors by Vpr is important for HIV-1 replication. However, the mechanisms mediating Vpr stability were unclear. Our data showed that PRMT5 and PRMT7 binding increase Vpr stability. We observed that virus production decreased PRMT5- and PRMT7-knockdown MDMs infected with HIV-1. These results showed that stabilized Vpr increased virus production at both the early (e.g., nuclear import) and late stages. In addition, because Vpr also suppressed antiviral responses [39], maintenance of Vpr protein stability may be critical for viral spread. Our results showed that low expression of PRMT5 and PRMT7 may destabilize Vpr protein, thereby suppressing viral spread. Therefore, low expression of PRMT5 and PRMT7 may contribute to increased resistance to HIV-1 pathogenesis.

In this study, we observed varying levels of expression of *PRMT5* and *PRMT7* mRNAs in native and siRNA-transfected MDMs, resulting in differing levels of viral productivity. This implied that wild-type HIV-1 cannot efficiently produce virus particles in a person with low PRMT5 or PRMT7 expression. In contrast, high expression of PRMT5 was not necessary for high viral production as evidenced by viral production in Donors 4 and 5 (high PRMT5 producers) being lower than in Donors 6 and 7 (low PRMT5 producers). In addition, the virus production in MDMs from Donor 3 showed high titer despite infection with Vpr-deficient strain. Based on these results, PRMT5 and PRMT7 indirectly support virus production. It is possible that high PRMT5 and PRMT7 expression levels may contribute to high viral loads in patient, while susceptible donors such as Donor 3 would not be affected by PRMT5 and PRMT7 expression level.

In conclusion, this study showed that expression of PRMT5 and PRMT7 assists virus production by modulating Vpr stability. However, further studies are required to reveal whether PRMT5 and PRMT7 expression is related to pathogenesis. Particularly, it is still unclear how PRMT5 and PRMT7 protect Vpr from proteasomal degradation. Thus, elucidation of the Vpr stabilization mechanism would reveal the specific conditions required for efficient virus production. In addition, further study is needed into how the expression levels of PRMT5 and PRMT7 contribute to the progression of a patient from HIV-1 infection to development of AIDS. 

## Figures and Tables

**Figure 1 viruses-12-00355-f001:**
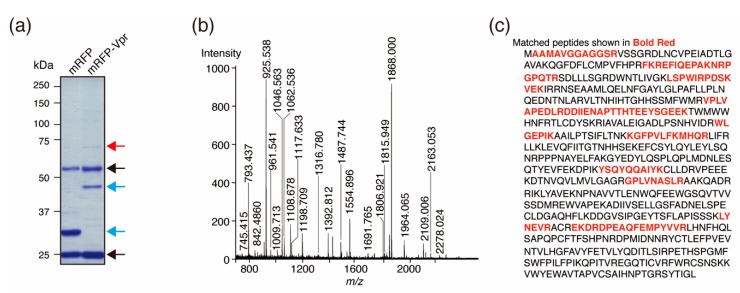
Identification of the Vpr-binding protein PRMT5. (**a**) Separation of Vpr-binding proteins by SDS-PAGE. The red arrow indicates the binding protein identified in this study. The blue arrows indicate FLAG-tagged-mRFP-FLAG-tagged-Vpr (mRFP-Vpr lane) and FLAG-tagged mRFP (mRFP lane). The black arrows indicate the heavy and light chains of the anti-FLAG IgG monoclonal antibody. (**b**) Peaks from MALDI-TOF MS analysis of the Vpr-binding protein after band excision and trypsin digestion. For peptide mass fingerprinting, the peak list was compared against the NCBI nr database using the MASCOT program. (**c**) The amino acid sequence of PRMT5 with matched peptide fragments identified by MALDI-TOF MS shown in red.

**Figure 2 viruses-12-00355-f002:**
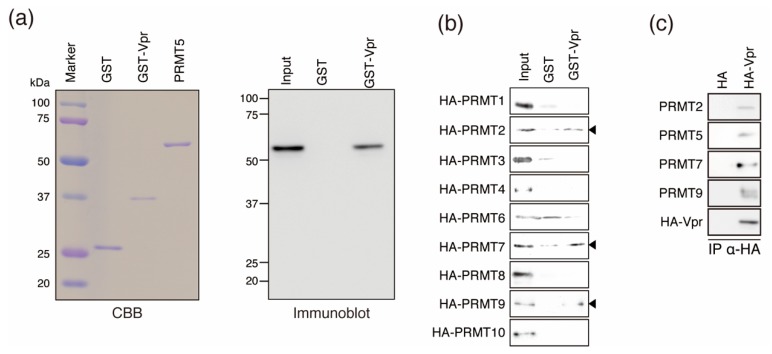
Binding of Vpr by PRMT family members. (**a**) GST pull-down assay using recombinant PRMT5. Purified GST, GST-Vpr, and PRMT5 expressed by *E. coli* strain BL21 cells were resolved by SDS-PAGE on 10% gels and stained with Coomassie brilliant blue (CBB; left panel). Recombinant PRMT5 after GST removal was incubated with GST-Vpr or GST as a control (immobilized on GST beads) and detected by immunoblotting with an anti-PRMT5 MAbs (right panel). (**b**) GST pull-down assays with other PRMT family members and GST-Vpr. Arrowheads indicate Vpr-binding PRMT proteins (PRMT2, PRMT7, and PRMT9). (**c**) Co-immunoprecipitation using HA-tagged-Vpr (HA-Vpr)-expressing cells. HA-Vpr and HA tag alone were expressed in HeLa cells and co-immunoprecipitated using anti-HA MAbs. PRMT2, PRMT5, PRMT7, and PRMT9 were detected by immunoblotting with antibodies specific to each protein.

**Figure 3 viruses-12-00355-f003:**
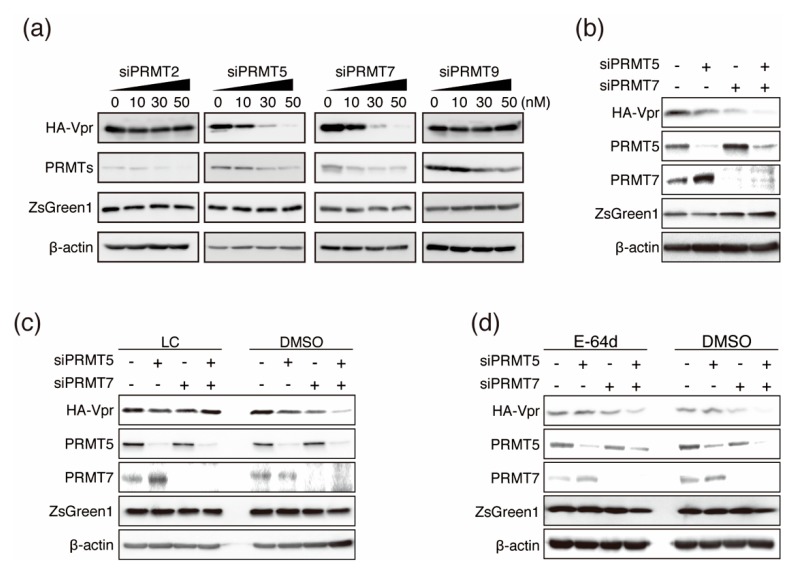
Downregulation of HA-Vpr after siRNA knockdown of PRMTs and rescue by inhibitors. (**a**) HA-Vpr expression using the bicistronic vector HA-Vpr-IRES-ZsGreen1-pCAGGS with siRNA against PRMT2, PRMT5, PRMT7, or PRMT9. Proteins were visualized by immunoblotting at 48 h post-transfection. Lanes marked “0” indicate transfection with siRandom as a negative control. (**b**) HeLa cells were co-transfected with HA-Vpr-IRES-ZsGreen1-pCAGGS together with siPRMT5 (50 nM), siPRMT7 (50 nM), or siRandom (50 nM) as indicated. Lanes marked “−” indicate transfection of siRandom. Proteins were visualized by immunoblotting at 48 h post-transfection. HeLa cells were cotransfected with HA-Vpr-IRES-ZsGreen1-pCAGGS and siRNAs against PRMT5 and/or PRMT7, and then treated with 50 µM lactacystin (LC) (**c**) or 50 µM E-64d (**d**) for 6 h starting at 48 h post-transfection. Proteins were visualized by immunoblotting.

**Figure 4 viruses-12-00355-f004:**
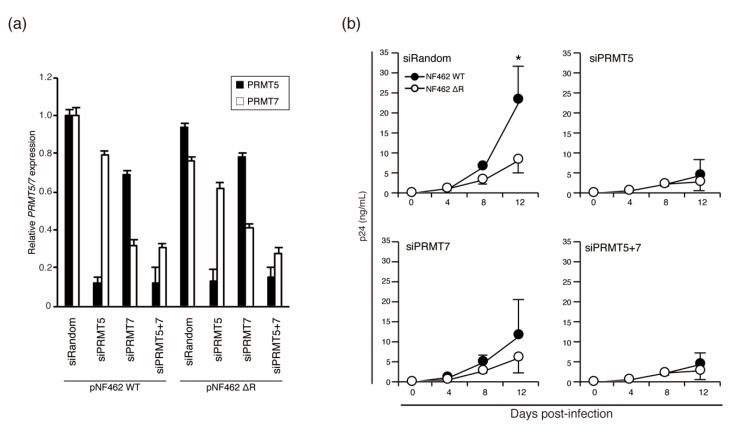
Knockdown of PRMT5 and PRMT7 in MDMs affects virus production. (**a**) Expression of *PRMT5* and *PRMT7* at 12 days post-infection was measured by qPCR. The experiments were independently conducted in triplicate. (**b**) MDMs were transfected with siPRMT5 and siPRMT7, and then infected with NF462 WT or NF462 ΔR strains of HIV-1. Virus production was measured using p24 ELISA at 0, 4, 8, and 12 days post-infection. Statistically significant differences are indicated by single asterisks (* *p* < 0.05).

**Figure 5 viruses-12-00355-f005:**
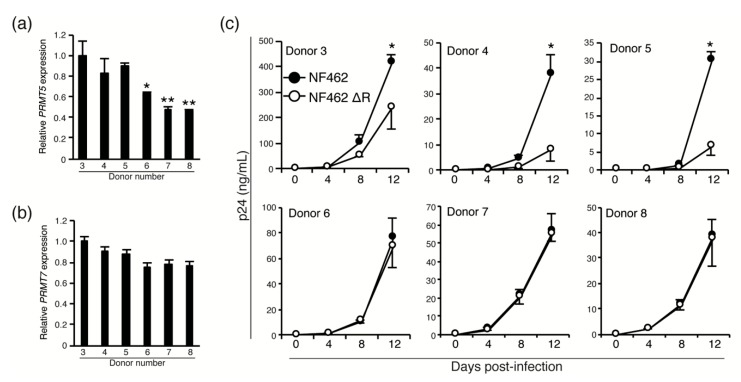
Differential expression of PRMT 5 in MDMs affected virus production. Expression of *PRMT5* (**a**) and *PRMT7* (**b**) mRNA in MDMs from six healthy donors. (**c**) MDMs from each donor were infected with the NF462 WT or NF462 ΔR strain, and virus production was measured using p24 ELISA at 0, 4, 8, and 12 days post-infection. Values shown are mean values, with error bars indicating standard deviations. Experiments were independently conducted in triplicate. Statistically significant differences are indicated by single and double asterisks (* *p* < 0.05 and ** *p* < 0.01, respectively).

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
