# Peer review of "Protein Arginine N-methyltransferases 5 and 7 Promote HIV-1 Production"

_viruses, 2020, doi:10.3390/v12030355_

Round 1

Reviewer 1 Report

Major comment #1: This comment has been sufficiently addressed.

Major comment #2: This comment has been sufficiently addressed.

Major comment #3: This comment has been sufficiently addressed.

Major comment #4: This comment has been sufficiently addressed.

Major comment #5: I still have concerns about the data in Figure 4. The authors measure knockdown by relative expression. On lines 365-370, the authors speculate that methylation of proteins may be involved. However, there is no evidence that this is the case. I would suggest removing this statement.

Major comment #6: I am still concerned about Figure 5. The authors say donors 3-5 has high PRMT5 levels compared with donors 6-8. First, the authors should use real time RT-PCR to quantify RNA levels with internal standards. It appears they are using donor #3 to determine the relative levels. Secondly, it is concerning that NF462ΔR infection of MDM from donor #3 produces higher levels of virus than NF462 (wild type virus) infection of MDM from donors 4-8. Third, the authors now state that protein levels of PRMT5 are 12 times lower than in HeLa cells, which suggests that it may not be involved in enhancing virus replication.

Major comment #7: In Figure 6, the authors attempt to correlate levels of PRMT5 and PRMT7 in patients with undetectable viral loads and those with viral loads. The authors now state the range of viral loads (67-160,000copies per ml). Several things concern me here. First, for those patients that had undetectable viral loads, was it due to cART or do these patients have naturally low HIV-1 viral loads? Secondly, how do these levels compare with healthy people? Third, the authors measured PRMT5 and 7 levels in monocytes. However, in the manuscript they analyze MDM, which may be different with respect to expression levels of PRMT5 and 7. Finally, it is concerning that half of the patients with no viral loads have PRMT5 levels that are comparable to those with viral loads. I don’t believe there is an association here and suggest removing this figure.

Major comment #8: Figure 7 was deleted. No concerns.

Author Response

Response to Reviewer 1’s comments

Major comment #5: I still have concerns about the data in Figure 4. The authors measure knockdown by relative expression.

Answer:

We estimated PRMT5 and PRMT7 expression using mRNA expression. The previous study demonstrated that the mRNA expression detected by qPCR correlated with protein expression level (Pal et al., EMBO J, 2007). In addition, recent study estimated the PRMT expression level using qPCR (Tan et al., BMC Cardiovasc Disord, 2019). The other recent study quantified mRNA expression of PRMT5 and PRMT7 normalized to that of beta-actin same as our method in order to quantify PRMT5 and PRMT7 expression level (Demetriadou et al., Cell Death Dis, 2019). Ideally, as Reviewer 1’s comment, the PRMT5 and PRMT7 expression should be demonstrated by protein detection, but we thought that our method using qPCR is reasonable for using small sample volume and low expression level. Therefore, we add explanation in “2.8. Quantitative PCR (qPCR)” section (lines 145-150) and “3.5. Knockdown of Vpr-binding PRMTs in MDMs inhibited efficient virus production” section (lines 262-265).

On lines 365-370, the authors speculate that methylation of proteins may be involved. However, there is no evidence that this is the case. I would suggest removing this statement.

Answer:

According to the Reviewer 1’s comment, our data did not show that the methylation relates Vpr stability. Therefore, we removed the statement in the revised manuscript. In addition, we added explanation that it is unknown that the protein methylation modified by PRMT5 and PRMT7 associated with its stability although the interaction between PRMT5/PRMT7 and Vpr would relates Vpr stability (third paragraph in “Discussion” section).

Major comment #6: I am still concerned about Figure 5. The authors say donors 3-5 has high PRMT5 levels compared with donors 6-8. First, the authors should use real time RT-PCR to quantify RNA levels with internal standards. It appears they are using donor #3 to determine the relative levels. Secondly, it is concerning that NF462ΔR infection of MDM from donor #3 produces higher levels of virus than NF462 (wild type virus) infection of MDM from donors 4-8. Third, the authors now state that protein levels of PRMT5 are 12 times lower than in HeLa cells, which suggests that it may not be involved in enhancing virus replication.

Answer:

First concern

I would like to apologize that the previous manuscript did not show internal standards. We quantified mRNA levels using internal standard, beta-actin. The revised manuscript shows that the beta-actin was used for an internal standard in this study. The method is same as the previous study’s method, as mentioned above.

Second concern

According to Reviewer 1’s comment, the virus production level in MDM from Donor 3 was higher other those from other donors. We think that the PRMT5 and PRMT7 function would be supportive but not on-off function for virus production, because Vpr-deficient strain-infected MDMs can produce virus particle. In addition, virus production was dependent on susceptibility of each MDM, and expressions of PRMT5 and PRMT7 could not affect susceptibility of MDMs from each donor. Therefore, the expression level of PRMT5 and PRMT7 would not be a central factor for susceptibility in each individual. However, the high expression level of PRMT5 could promote virus production in wild-type strain by exertion of Vpr function because the PRMT5 knockdown resulted in low virus productivity of wild-type strain like that of Vpr-deficient strain. The low expression level of PRMT5 may spoil virus productivity of wild-type strain like that of Vpr-deficient strain because PRMT5 cannot protect Vpr. By contrast, in MDMs from Donor 3, MDMs with high susceptibility would not be affected by PRMT5 expression level. Therefore, we modified discussion section according to Reviewer 2’s comments (second and fifth paragraph in “Discussion section).

Third concern

I would like to apologize that our explanation was not enough. To explain why the PRMT proteins from MDMs was not clearly detected by immunoblotting, we compared PRMT mRNA level between MDMs and HeLa cells. The experiment showed suitable detection method which the PRMT5 and PRMT7 expression levels were detected by qPCR because previous study demonstrated that PRMT mRNA expression was correlated with protein expression, as mentioned above.

According to the Reviewer 1’s comment, it was not suitable statement, which the PRMTs associated with promoting virus replication, based on data shown in only Figure 5. However, the data showed in Figure 4 support the data shown in Figure 5. Therefore, we added explanation in “Discussion” section (first paragraph).

Major comment #7: In Figure 6, the authors attempt to correlate levels of PRMT5 and PRMT7 in patients with undetectable viral loads and those with viral loads. The authors now state the range of viral loads (67- 160,000copies per ml). Several things concern me here. First, for those patients that had undetectable viral loads, was it due to cART or do these patients have naturally low HIV-1 viral loads? Secondly, how do these levels compare with healthy people? Third, the authors measured PRMT5 and 7 levels in monocytes. However, in the manuscript they analyze MDM, which may be different with respect to expression levels of PRMT5 and 7. Finally, it is concerning that half of the patients with no viral loads have PRMT5 levels that are comparable to those with viral loads. I don’t believe there is an association here and suggest removing this figure.

Answer:

According to Reviewer 1’s comments, we modified discussion that the PRMT5 would not be a central factor for individual difference. The revised conclusion could not support data shown in Figure 6 in the previous manuscript. Therefore, we removed the Figure 6 data.

Reviewer 2 Report

I would still recommend another round of English editing, as there are many grammatical mistakes throughout the manuscript (see e.g. new line 73: ‘The human PBMCs were collected eight different healthy donors’).

Other comments:

- Lines 65-66: ‘all PRMT proteins, except PRMT8, are expressed in all tissues’ seems a bit of an odd statement, as the authors continue by saying that macrophages have not been tested. Thus, twice using ‘all’ is not appropriate. Please reword

- I assume that the HIV-1 infected patients were not treated with antiretroviral therapy?

- lines 356-362: the words different/differed/difference are used eight times here. Please rewrite

- I could not assess the supplementary information, as the files were somehow not available

Author Response

Response to Reviewer 2’s comments

I would still recommend another round of English editing, as there are many grammatical mistakes throughout the manuscript (see e.g. new line 73: ‘The human PBMCs were collected eight different healthy donors’).

Answer:

According to the Reviewer 2’s comment, the current manuscript has been proofread by Editage and this has been added in “Acknowledgments” section.

The sentence pointed out by Reviewer 1 was revised as ‘The human PBMCs were collected “from” eight different healthy donors’ (line 71).

Other comments:

- Lines 65-66: ‘all PRMT proteins, except PRMT8, are expressed in all tissues’ seems a bit of an odd statement, as the authors continue by saying that macrophages have not been tested. Thus, twice using ‘all’ is not appropriate. Please reword

Answer:

We replaced “in all tissues” with “in various tissues” (line 65).

- I assume that the HIV-1 infected patients were not treated with antiretroviral therapy?

Answer:

The Figure 6 was removed according to the Reviewer 1’s comment.

- lines 356-362: the words different/differed/difference are used eight times here. Please rewrite

Answer:

According to the Reviewer 2’s comment, we revised the repetitious words (second paragraph in “Discussion” section).

- I could not assess the supplementary information, as the files were somehow not available

Answer:

I resubmitted revised supplementary data.

Round 2

Reviewer 1 Report

I have reviewed the above revised manuscript and I am fine with the authors changes.

This manuscript is a resubmission of an earlier submission. The following is a list of the peer review reports and author responses from that submission.

Round 1

Reviewer 1 Report

The manuscript by Murakami et al concerns the analysis of host proteins able to bind HIV-1 Vpr in monocytes and macrophages. The authors identify two such proteins, PRMT5 and PRMT7, whose expression levels correlate with virus production in cells of the monocytic lineage. They hypothesize that these two proteins stabilize Vpr by binding to it.

Most of the results shown are convincing, but I have some problems, especially with Figs. 5 and 7.

Fig. 5: Natural variation in PMRT5 and -7 expression in donors is correlated here with HIV-1 replication. The authors use two viruses, a wt virus, and one deficient in Vpr. This latter virus, NF462deltaR, has not been described in detail. The authors refer to ref. 43, with a further reference to Iijima et al (2004). In that paper, only a R8788EA mutant has been described in which the amino acid mutated somewhat resembles the deltaR variant tested here. Because it is important to use a non-functional protein as a control, the authors should describe their mutant in more detail, also to the Vpr function(s) it lacks. In panels A and B, the Y-axis should be labelled PMRT5 and PMRT7, respectively (same for Fig. 7). Furthermore, the same scale should be used for the Y-axis in panel C. But now for the data in Fig. 5: in donors 4-6, with significantly lower expression of PMRT5 (Panel A), both the wt and the mutant virus replicate to similar levels as in donors 2 and 3, who have much higher expression of PMRT5. So, the results here contradict the hypothesis posed by the authors. Please explain.

In Fig. 7, lymphoid cell lines are analysed as well, and it is obvious that in Jurkat T cells, the expression of PMRT5 and -7 is associated with latency similar to the situation in myeloid cell lines. Would this imply that PMRT5 and -7 also play a role in lymphoid cells infected with HIV, refuting the claim made by the authors that these proteins are specifically involved in infected monocytes and macrophages only? (See also Discussion, lines 358-360, for claims refuted by Figs. 5 and 7). Next, I wonder why PMRT5 and -7 expression is lower in latently infected cell lines than in their parent lineages (Fig. 7)? The authors write about ‘congenital’ expression in the donors used, but it appears here that expression level of these two proteins is affected by HIV-1 infection. I do not suppose that the latently infected lineages have any novel mutations in the promoter regions of the PMRT5 and -7 genes? If expression of PMRT5 and -7 can be down-regulated by HIV-1 infection, then I presume it can also be upregulated? For instance by wtVpr? Or certain Vpr variants? Which could likewise explain virus production levels. Maybe some viruses are better in upregulating PMRT5 and -7 expression? Could the authors commend on this alternative explanation?

Minor remarks

- The English usage and grammar should be thoroughly checked throughout the manuscript (line 61-62: carries out its functions are exerted; line 198; Vpr protein at C-terminal was encoded arginine rich sequence; line 297: ‘spectrum’ is used twice; and at many other places).

- line 172: what does ‘subsection’ mean here?

- line 195-196: Was it to be expected that the endogenous expression level would change?

- line 200: explain why a ‘potential methylation site’ would be important.

- line 203: Suppl. Fig. 2: what does ‘cayno’ mean in the name?

- line 203-205: the authors state that there is ‘no direct interaction’ between Vpr and the identified proteins. But they bind each other, right? That would be a direct interaction.

- line 227: explain ‘induced the result of proteasomal degradation’ as unclear to me what is meant.

- line 232: which donor was used here?

- line 252: did not differ significantly?

- Fig. 1: panels B and C: what are the peaks not indicated in red?

- Fig. 3: panel A: beta-actin expression is not the same between conditions.

- lines 377: only in monocytes? Or also in T cells?

Reviewer 2 Report

The manuscript by Murakami and colleagues describe the interactions of the HIV-1 Vpr protein with protein arginine methyltransferases 5 and 7. The manuscript is generally well written; however there concerns about the validation of the experimental approach, the lack of controls in some experiments and the conclusions drawn from some of the results. I have difficulty accepting that HIV-1 replication in monocyte derived macrophages (MDM) can be so drastically different by less than two-fold levels of PRMT5.  

Major comments:

1)    Figure 1: Vpr is known to interact with several proteins including DCAF1, Sap145, p300, SLX4 and α-importin. In Fig. 1, the authors show an interaction with a protein that was shown to be PRMT5 by MADLI-TOF.  Examination of the gel reveals the presence of several other bands. If the authors should identify these proteins and determine if they proteins previously found to interact with Vpr. If so, it would validate the approach.  

2)    Figure 2: In Fig. 2a (right side panel), the authors show the results of incubating GST or GST-Vpr with PRMT5 followed by immunoblotting with an antiPRMT5 antibody. While the GST-Vpr appears to co-immunoprecipitated with PRMT5, the GST lane also shows some reactivity. This is somewhat disturbing and needs to be explained.

3)    Figure 2: In Fig 2c, the authors show GST pulldowns of PRMT2, 5, 7, and 9 using HA-Vpr. However, the authors need to include an unrelated protein control tagged with HA.  Also, the HA control appears to pulldown PRMT2.  Why??

4)    Figure 3: The authors attempt to show knockdown of PRMT2, 5, 7, and 9 using siRNA. The authors state, “Interestingly, the knockdown of PRMT5 or PRMT7 downregulated the expression of HA-Vpr in a dose-dependent manner, but did not affect expression of ZsGreen1; however, knockdown of PRMT2 and PRMT9 did not affect expression of HA-Vpr and ZsGreen1 (Figure 3a).”  This is not correct as the level of PRMT5 and PRMT7 appears to be similar from 10nM to 50nm.  This needs to be corrected. The authors need to quantify the levels on the immunoblots. Also, the siRNA known down of PRMT2 and 9 does not appear to be robust and therefore the authors can’t really say it influenced HA-Vpr stability. Additionally, the known down of PRMT5 does not appear to lead to much of a degradation of HA-Vpr from 10-50nM and with knock down of PRMT7 there is only a decrease in HA-Vpr at the 50nM concentration while PRMT7 appears to be at the same level.

5)    Figure 4: In this figure the authors examine HIV-1 and HIV-1Δvpr in the presence of siRNA against PRMT5, PRMT7 or PRMT5+7. The siRNA against PRMT5 clearly reduces virus production of both HIV-1 and HIV-1Δvpr while PRMT7 has a minimal effect. PRMT5 participates in various biological processes, such as transcriptional regulation, ribosome biogenesis, RNA metabolism, and cell cycle regulation. So one has to ask whether the downregulation of PRMT affect these processes and is the reason for decreased virus replication? This should be addressed by the authors.

6)    Figure 5: In this figure, the authors examine the mRNA levels of PRMT5 and 7 from six healthy donors.  In Fig. 5A, the authors show that 3 donors (1-3) had higher mRNA levels of PRMT5 than the other 3 donors (4-6). The authors state that the mRNA differences between donors 1-3 and 4-6 were statistically significant while the levels of mRNA for PRMT7 were not significantly different. They correlate these findings with the ability of HIV-1 and an HIV-1Δvpr variant to replicate in MDM.  With groups 4-6, HIV-1 and HIV-1 Δvpr replicated the same despite a 2-fold or less difference in mRNA. There are some concerns with the findings.  First, it is dangerous to make statements based on this number of samples. Second, with donors 1-3, which similar levels of PRMT5, the differences in replication were ~2-6 fold. Third, the authors need to examine the MDM for protein expression to confirm the qPCR results.  

7)    Figure 6: In this figure the authors compare the mRNA levels of PRMT5 and 7 in HIV-1 patients with undetectable viral loads (8 patients) and those with viral loads (9 patients). With PRMT5, PRMT5 mRNA levels were very low in all 9 patients with detectable loads while those with viral loads there was a greater range in mRNA levels. However, more than half the patients with viral loads had PRMT5 levels comparable to the undetectable viral load patients. Similar results were obtained with PRMT7.  There are several concerns here that need to be addressed. First, in the population with detectable viral loads, what were the viral loads?  Second, was there a correlation between the viral loads and mRNA levels of PRMT5 and 7. Third, were the patients with higher mRNA levels of PRMT5 also the patients that had higher mRNA levels of PRMT7?

8)    Figure 7: In this figure the authors examine four myeloid and four lymphoid cell lines for PRMT5 and PRMT7. Two of each type of cell lines are latently infected with HIV-1. The authors state that with the myeloid cell lines PRMT5 is more important for virus production than PRMT7. There are several problems with this experiment and accompanying statements. First, they only examined two latently infected and two non-infected cell lines and it is hard to draw strong conclusions based on two cell lines. Secondly, the difference in PRMT5 RNA levels was less than two-fold and whether this is biologically relevant is uncertain. Third, the authors need to analyze the levels of PRMT5 protein from these cell lines. Finally, the authors can’t state that the presence of PRMT5 is more important for virus production unless they perform growth curves in these cell lines.

Minor comments:

1)    Line 52-53: In the sentence, “In addition, HIV-1-infected macrophages maintain viremia,” should be changed to, “In addition, HIV-1-infected macrophages contribute to viremia.” It is currently unknown (and improbable) that discontinuation of HAART leads to viremia exclusively from infected macrophages.

2)    Line 66-67: This sentence does not make sense and should be reworked: “We found that two protein arginine 67 N-methyltransferases (PRMTs), PRMT5 and PRMT7, could bind Vpr and related to virus production.”

3)    Line 102: In the sentence: The Monocytes… should be changed to: The monocytes…